# Extended stop codon context predicts nonsense codon readthrough efficiency in human cells

Kotchaphorn Mangkalaphiban[1,2], Lianwu Fu[3], Ming Du[3], Kari Thrasher [3], Kim M. Keeling [3], David M. Bedwell [3] & Allan Jacobson [1]

Protein synthesis terminates when a stop codon enters the ribosome's A-site. Although termination is efficient, stop codon readthrough can occur when a near-cognate tRNA outcompetes release factors during decoding. Seeking to understand readthrough regulation we used a machine learning approach to analyze readthrough efficiency data from published HEK293T ribosome profiling experiments and compared it to comparable yeast experiments. We obtained evidence for the conservation of identities of the stop codon, its context, and 3'-UTR length (when termination is compromised), but not the P-site codon, suggesting a P-site tRNA role in readthrough regulation. Models trained on data from cells treated with the readthrough-promoting drug, G418, accurately predicted readthrough of premature termination codons arising from CFTR nonsense alleles that cause cystic fibrosis. This predictive ability has the potential to aid development of nonsense suppression therapies by predicting a patient's likelihood of improvement in response to drugs given their nonsense mutation sequence context.

Termination of protein synthesis occurs when the ribosome encounters one of the three stop codons (UAA, UAG, and UGA) at the end of an mRNA open reading frame (ORF). The release factor complex, comprised of eRF1, eRF3, and GTP, recognizes a stop codon in the ribosomal A-site and facilitates nascent peptide release, then the ribosome is recycled by Rli1/ABCE1 for another round of translation[1,2]. The termination process is highly efficient and has a low error rate[3], but an error can still occur when the stop codon is decoded by a near-cognate tRNA instead of eRF1, resulting in continued translation elongation into the mRNA 3'-UTR, a process termed "stop codon readthrough"[3,4]. An understanding of the details of this process is likely to be beneficial for the development of therapeutics that target diseases caused by nonsense mutations, where readthrough is desirable at premature termination codons (PTCs) but not normal termination codons (NTCs)[5].

*Cis*-acting elements influencing the efficiency of termination and readthrough have been identified in several organisms. These modulators of readthrough efficiency include the identity of the stop codon and flanking nucleotides[6–10], stem–loop structures in the mRNA 3'-UTR[6,11], and specific RNA binding protein motifs[12,13]. Among these, the stop codon and the nucleotide following it (nt +4) are the most studied across species and yield the most consistent results. Structural insights into their mechanism have been demonstrated with cryo-electron microscopy[14,15] and their significance has been extended to a transcriptome-wide level in human and yeast cells by ribosome profiling experiments[16,17]. The conservation of other elements, however, is more difficult to determine due to variations in experimental conditions, analysis strategies, and (for endogenous mRNAs) existing nucleotide usage biases in different species. The diversity in mRNA sequences in transcriptomics data provides opportunities to

[1]Department of Microbiology and Physiological Systems, UMass Chan Medical School, 368 Plantation Street, Worcester, MA 01655, USA. [2]Department of Genomics and Computational Biology, UMass Chan Medical School, 368 Plantation Street, Worcester, MA 01655, USA. [3]Department of Biochemistry and Molecular Genetics, Heersink School of Medicine, The University of Alabama at Birmingham, 845 19th Street South, Birmingham, AL 35294, USA. e-mail: allan.jacobson@umassmed.edu

determine whether *cis*-acting elements are conserved for endogenous mRNAs between yeast and human cells, as well as to explore how they can be utilized to predict readthrough efficiency given new sequences.

Therefore, to gain parallel insights into *cis*-acting elements affecting readthrough efficiency at a human transcriptome-wide level, we re-analyzed published readthrough efficiency data for HEK293T cells[17] in the same manner as we did for yeast cells[16]. Our analysis revealed that, in addition to the previously established importance of stop codon context, readthrough efficiency increased with 3′-UTR length in HEK293T cells under readthrough-promoting conditions, consistent with our yeast results. We also found that the patterns of P-site codon triplets associated with high or low readthrough efficiency in HEK293T cells are largely different from those observed in yeast. Further, we demonstrated that a machine learning model trained on these *cis*-acting elements can predict with high accuracy the readthrough efficiency of PTCs arising from nonsense mutations found in cystic fibrosis patients. Collectively, we obtained evidence for the conservation of *cis*-acting elements modulating readthrough efficiency among human and yeast cells at a transcriptome-wide level, derived insights into the mechanism of translation termination that may involve tRNA properties, and presented potential applications of ribosome profiling data coupled with machine learning approaches in readthrough prediction and nonsense suppression therapies.

## Results

### Random forest models identify mRNA features predictive of readthrough efficiency

To study *cis*-acting elements affecting transcriptome-wide readthrough efficiency in human cells, we applied the analysis approaches we previously developed with yeast ribosome profiling data[16] to readthrough efficiency data generated from ribosome profiling experiments of HEK293T cells treated with various readthrough-promoting aminoglycosides[17]. For each sample, mRNAs with detectable readthrough were identified, and random forest models[18,19] were trained to predict the readthrough efficiency of these mRNAs using mRNA or nascent peptide features (Fig. 1, Supplementary Fig. 1). The feature importance score of each feature extracted from the model indicates the predictive ability of that feature (Fig. 1). As a negative control for baseline unimportant feature, we randomly assigned a number (1–100) and a letter (A, C, G, U) to each mRNA, both of which were expected to have no roles in readthrough efficiency prediction. Indeed, the two negative control features have low feature importance scores in all samples for both random forest approaches: the regression models trained to predict readthrough efficiency values directly (Fig. 1a, negative control = "NC" columns) and the classification models trained to predict extremely "high" and "low" readthrough groups (top and bottom 15%) (Fig. 1b, negative control = "NC" columns). Among the

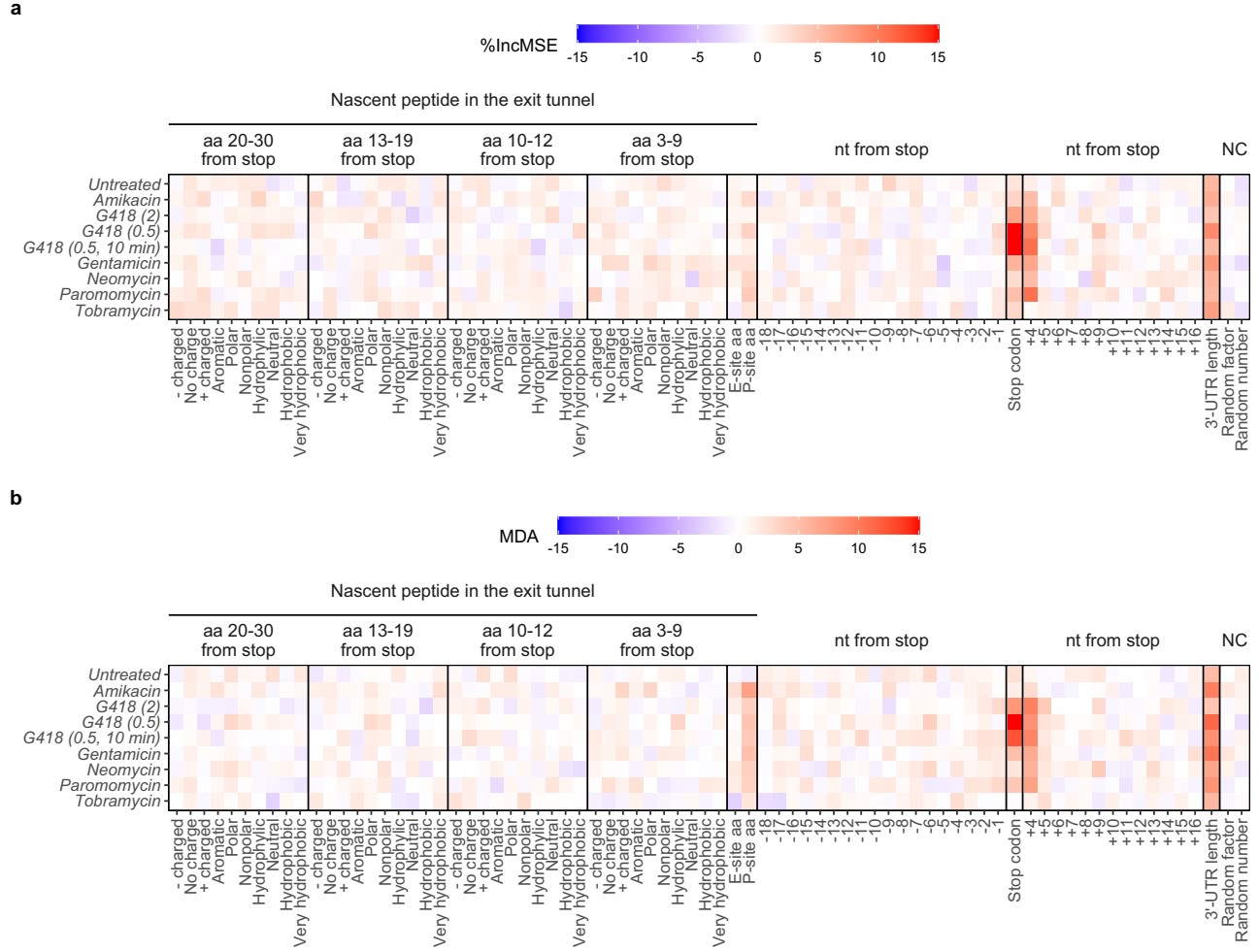

**Fig. 1 | mRNA features predictive of readthrough efficiency.** Feature importance scores, % increase in mean square error (%IncMSE) for regression (**a**), and mean decrease accuracy (MDA) for classification (**b**) were extracted from random forest models. The higher the feature importance score (red), the more important a feature is in predicting readthrough efficiency or distinguishing between "high" and "low" readthrough mRNAs. Numbers or categorical values randomly assigned to the mRNAs are used as negative controls ("NC") and to set the baseline feature importance scores as unimportant features. aa amino acid, nt nucleotide. Source data are provided as a Source Data file.

features with high scores that are predictive of readthrough efficiency are the identities of the stop codon and the nucleotide immediately after it (nt +4) (Fig. 1), each of which has previously been shown to affect termination and readthrough efficiency[3,4,20]. Other features with high scores include 3'-UTR length and the identity of P-site amino acid (Fig. 1), both of which were previously recognized in yeast[16,21]. For the stop codon and nt +4, the feature importance scores for both are especially prominent in G418-treated samples, where readthrough levels were much higher than those seen in HEK293T cells treated with other aminoglycosides and 3'-UTR ribosome footprints showed 3-nt periodicity indicative of continued translation into the 3'-UTR from the coding (CDS) region[17].

## Known stop codon context influences readthrough efficiency

To understand how mRNA features affect readthrough efficiency prediction, we grouped mRNAs based on the usage of stop codons, nt +4, or other nucleotides flanking the stop codons and compared the median readthrough efficiency of that group to the sample median (Fig. 2a). When mRNAs were grouped by the identity of the negative control random letters, no significant differences between sample median and each group's median were observed in any samples (Fig. 2a, "Random"). However, when mRNAs were grouped by the stop codon identity, we found that mRNAs that used UGA as the stop codon had higher readthrough efficiency, while those using UAG and UAA had lower readthrough efficiency than the overall sample median in

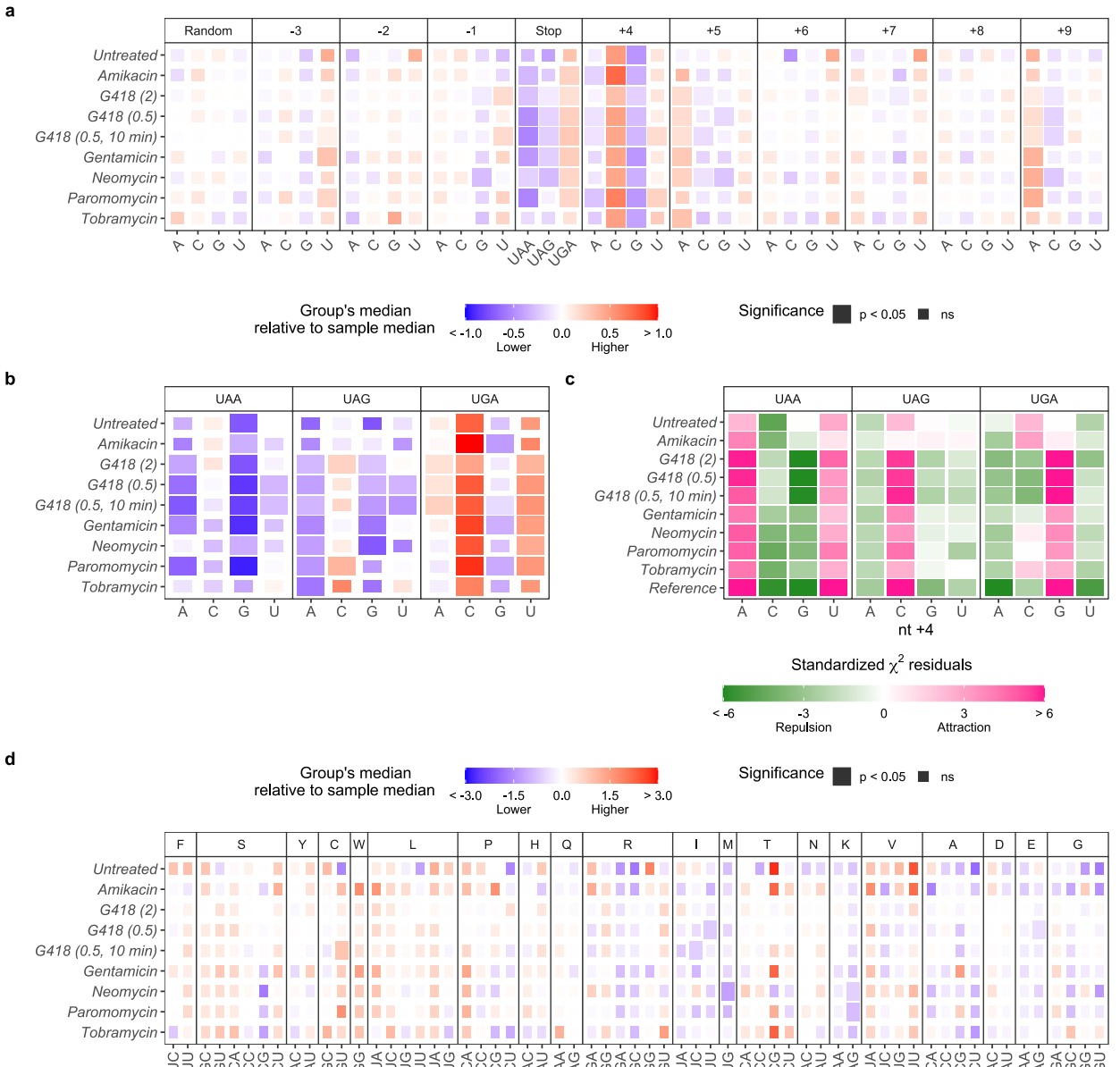

**Fig. 2 | Effects of the stop codon and flanking nucleotide identities on readthrough efficiency. a, b, d** Differences between median readthrough efficiency of all mRNAs in the sample and median readthrough efficiency of a group of mRNAs containing particular stop codon or nucleotide (**a**), stop codon with nt +4 as quadruplet (**b**), or triplet codon in the ribosomal P-site (**d**). Positive (red) and negative (blue) values indicate that the group of mRNAs had higher and lower median readthrough efficiency compared to the sample median, respectively. Two-tailed Wilcoxon's rank sum test with the Benjamini–Hochberg method for multiple testing correction was used to determine whether the difference was significant. **c** Standardized residuals of two-tailed $\chi^2$ test of independence determining association between stop codon and nt +4 identities. Positive residuals (pink) indicate that the pair occurs together more often than expected (attraction), while negative residuals (green) are less often than expected (repulsion). For all panels, a significant result ($p < 0.05$) is represented as a larger tile. Source data, the exact $p$-values, and the number of data points ($n$) in each group are provided as a Source Data file.

most samples (Fig. 2a, "Stop"). This observation is consistent with results from Wangen and Green despite the difference in statistical analysis approaches, as well as with results from previous studies showing that UGA is the most readthrough-permissive stop codon, while UAA is the least[3,4,7,22]. Using our approach, we can extrapolate from the general trends across samples that the readthrough-promoting motif is UGACANNNA while the readthrough-inhibitory motif is UAAG(G/C)NNNC in these HEK293T cells.

Currently, there are 30 mRNAs with experimentally validated programmed stop codon readthrough (SCR)[13,23], many of which have a readthrough motif of UGACUAG. We determined whether these known SCR mRNAs have higher readthrough efficiency in the ribosome profiling data set analyzed here. While not all SCR mRNAs are expressed or have sufficient expression for their readthrough to be detected, those that can be detected (9 out of 30 across samples) tend to have higher readthrough efficiency than average (Supplementary Fig. 2a, red and green). Since 29 out of the 30 readthrough mRNAs (all 9 detected here) use UGA stop codons, we compared the readthrough of these mRNAs against the distribution of readthrough efficiency of all UGA-containing mRNAs and reached a similar conclusion (Supplementary Fig. 2b, red and green). The observations for UGACUAG (Supplementary Fig. 2b, red) are also consistent with the association of the UGAC quadruplet with high readthrough. However, there are a few SCR mRNAs that exhibited lower readthrough than average, suggesting that other mRNA features also play a role in regulating readthrough efficiency.

We further asked whether the strongest features, the stop codon and nt +4, together have additive effects on readthrough efficiency, as shown in previous studies. As expected, when mRNAs were grouped by the combination of these two features, the most readthrough-permissive combination, UGAC, had the highest increase in readthrough efficiency relative to the sample median, while the most readthrough-inhibiting combination, UAAG, had the highest decrease in readthrough efficiency across most samples (Fig. 2b). The trends for each feature when another feature is held constant (Fig. 2b) are also mostly in line with analyses performed for each feature independently (Fig. 2a). When nt +4 is held constant, the order of most to least readthrough-promoting stop codon is UGA > UAG > UAA (Fig. 2b, compare "A" columns from different stop codon panels to each other, and so on). When the stop codon is held constant, the order most to least readthrough-promoting nt +4 is C > U > A/G (Fig. 2b, compare different nt +4 columns to each other within a specific stop codon panel).

## A readthrough-promoting stop codon and nt +4 combination appears to be selected against throughout evolution

Unlike studies of readthrough using reporters, studying readthrough of endogenous mRNAs can be affected by the biased frequency of codon or nucleotide usage that results from various evolutionary pressures. On the one hand, the existence of programmed readthrough genes with readthrough permissive contexts suggests that readthrough is allowed to happen to facilitate protein evolution[13,24]. On the other hand, the fact that the stop codon with the highest fidelity, UAA, is concentrated among mRNAs of highly expressed and essential genes suggests that readthrough is selected against to limit deleterious consequences of a C-terminally extended polypeptide[7,25–28]. Studying readthrough of endogenous mRNAs using ribosome profiling further biases the pool of data due to a detection threshold for 3'-UTR reads, which depends on: (i) sufficiently high expression and translation of the mRNA CDS region, (ii) the existence of readthrough-promoting features that allow readthrough of certain mRNAs to be seen more often, and iii) aminoglycoside treatment (Supplementary Fig. 3a). Indeed, we observed that, on average, ~54% of mRNAs in each sample had UGA stop codons; these frequencies are only slightly higher than the 49% determined from the Reference group, which encompasses

mRNAs from all samples combined regardless of whether readthrough is detected or not (Supplementary Fig. 3b). For nt +4, A and G are the most common with comparable frequencies, followed closely by C, while U is depleted (Supplementary Fig. 3b). The high abundance of readthrough-promoting UGA but depletion in readthrough-promoting nt +4 suggests that a combination of readthrough-promoting features may have been under negative selection.

To statistically determine if certain pairs of stop codon and nt +4 are over- or under-represented given existing biases in the data, we performed a $\chi^2$ test of independence (Fig. 2c), comparing the observed (Supplementary Fig. 3c, d) to the expected equal frequencies of nt +4 among stop codons (Supplementary Fig. 3e, f). We found that UGAG is over-represented in all samples except Untreated, Amikacin-treated, and Tobramycin-treated samples, where UGAC is over-represented instead (Fig. 2c, pink in UGA panel). Coincidentally, these three samples have the fewest mRNAs with detectable readthrough (Supplementary Fig. 3a). While UGAG is still a very common quadruplet, UGAC becomes almost as common in the three samples (Supplementary Fig. 3c). These results indicate that when overall readthrough levels are very low, detectable readthrough in Untreated, Amikacin-treated, and Tobramycin-treated samples are in part biased towards mRNAs with readthrough-promoting features. Among the aminoglycoside-treated samples, G418-treated samples show the closest results to the Reference (Fig. 2c), indicating that readthrough induced by G418 led to a high enough number of mRNAs with detectable readthrough (Supplementary Fig. 3a) such that the diversity of stop codon and nt +4 combinations is sufficiently representative of the pool of mRNAs expressed in HEK293T cells. We focused on the Reference and G418-treated samples in further discussion of $\chi^2$ test results.

As described above, we determined that UGAG being noticeably the most common quadruplet in the data (Supplementary Fig. 3c) is not solely due to the naturally high abundance of UGA and G in the data, but their occurrence together is also more frequent than what would be expected by random chance of two highly abundant features (Fig. 2c). The same is true for UAAA being the most common among the UAA and nt +4 combinations (Supplementary Fig. 3c, d and Fig. 2c). These observations suggest that a combination of readthrough-promoting features tend to be selected against.

It is important to note that a combination being over-represented does not equate to their abundance becoming the highest. Although UAAU also occurs more often than expected (Fig. 2c), the low prevalence of U in general results in UAAU being relatively infrequent overall, not that much more frequent than UAAC or UAAG (Supplementary Fig. 3c). Similarly, although UAGC occurs more often than expected (Fig. 2c), its frequency is no more than UAGA or UAGG (Supplementary Fig. 3c). Because $\chi^2$ analysis compares proportions, it is expected that when G is over-represented in one group, other nucleotides like U or C would be under-represented in that group and would appear over-represented in another group. This is likely the reason for the positive associations seen for UAGC and UAAU rather than them being evolutionarily selected because their actual frequencies in the data are quite low (Supplementary Fig. 3c).

Taken together, we found that UGAG and UAAA occur more often than expected. This diminished association between readthrough-promoting features is likely the consequence of evolutionary pressure to limit high readthrough among mRNAs expressed in HEK293T cells.

## P-site codons have mild effects on readthrough efficiency

Some nucleotide positions upstream of the stop codon seem to be associated with different readthrough efficiency levels. For example, mRNAs with guanine at position −1 and those with uracil at position −3 had lower and higher readthrough efficiencies than the sample median, respectively (Fig. 2a). These nucleotide positions encode the amino acid immediately prior to the stop codon, so when the stop codon enters the ribosomal A-site, these nucleotides are in the P-site.

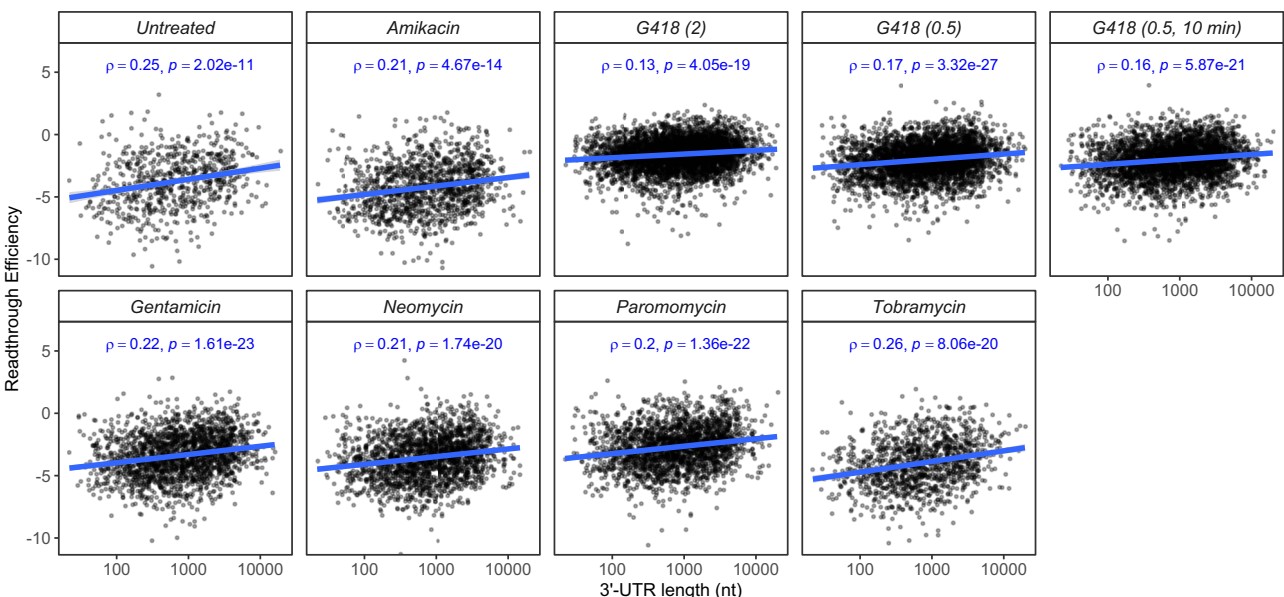

**Fig. 3 | Readthrough efficiency increases with 3′-UTR length.** Readthrough efficiency vs. 3′-UTR length for all mRNAs (that have UTR annotations) in the sample. Two-tailed Spearman's correlation coefficient (ρ) and the associated *p*-value are reported for each sample. Source data are provided as a Source Data file.

Concurrently, the P-site amino acid has feature importance scores higher than the baseline in the random forest models (Fig. 1).

To determine whether readthrough efficiency is modulated by the P-site codon nucleotides, encoded amino acids, or decoding tRNAs, we grouped mRNAs by the identity of their P-site codon triplets and compared each group's median readthrough efficiency to the sample median (Fig. 2d). If readthrough efficiency is strongly influenced by the nucleotides, we expect the codons with the same nucleotide composition at certain positions to have the same results regardless of the amino acid they encode. On the other hand, if readthrough efficiency is influenced by the amino acid, we expect the codons that encode the same amino acid to have the same results despite their differences in nucleotide composition. Results that do not follow these expectations suggest that tRNA properties may be involved or indicate a more complex interplay of at least two of the three P-site features.

Although no codons show significant results consistently across samples, the direction of association remains uniform across samples for many codons, as opposed to non-uniform patterns of negative control random letters (Fig. 2a, "Random"). For example, mRNAs that have AAG in the P-site generally have lower readthrough efficiency than the sample median, while those that have UGG have higher readthrough efficiency than the sample median (Fig. 2d). Because UGG is the only codon encoding tryptophan (W), it is not discernable whether this specific codon nucleotide combination, the decoding tryptophan tRNA, or the encoded tryptophan is responsible for higher readthrough efficiency. It is notable, however, that the first nucleotide of UGG corresponds to U at position −3, which is associated with high readthrough efficiency (Fig. 2a), and most other codons that begin with U also show the general trend of higher readthrough efficiency (regardless of significance level) (Fig. 2d). Therefore, U at position −3 may be mechanistically responsible for high readthrough efficiency in these mRNAs.

### Readthrough efficiency increases with 3′-UTR length
Random forest models identified 3′-UTR length as one of the important predictors of readthrough efficiency. To understand the relationship between 3′-UTR length and readthrough efficiency, we calculated Spearman's correlation coefficient (ρ) between these two values.

Readthrough efficiency is positively correlated with 3′-UTR length in all samples (Fig. 3). To rule out the possibility that the observed correlation is skewed by extremely short or long 3′-UTRs, we repeated the analysis with only those mRNAs whose 3′-UTRs are longer than 100 nt and shorter than 5000 nt. The positive correlations are maintained in all samples (Supplementary Fig. 4a). The positive correlation between readthrough efficiency and 3′-UTR length is probably linked to the role of poly(A)-binding protein (PABP) in enhancing translation termination, which has been demonstrated both in vivo and in vitro in other eukaryotes[21,29]. Further, in a reporter gene assay, the distance of PABP to the stop codon also correlated with readthrough efficiency in yeast[21].

It is possible that the 3′-UTR effect reflects an involvement of mRNA secondary structure in the 3′-UTR, which has been shown to induce readthrough in some genes. To test whether the observed 3′-UTR effect is related to mRNA secondary structure, we used an established prediction tool, RNALfold of the ViennaRNA package[30], to identify potential structures in the 3′-UTRs and calculated pairwise Spearman's correlation coefficient (ρ) between the minimum free energy (MFE) of the most stable predicted structure for a given 3′-UTR sequence (lowest MFE), readthrough efficiency, and 3′-UTR length (Supplementary Fig. 5a). The majority of 3′-UTRs (except for 7 across all samples) have at least one predicted structure. We observed that in all samples, the relationship between readthrough efficiency and MFE is much weaker than that between readthrough efficiency and 3′-UTR length; in some cases, there is no correlation at all. However, the strongest correlation is the 3′-UTR length vs. MFE; the negative correlation indicates that the longer the 3′-UTR is, the lower the MFE is (i.e., the stronger the structure is). These results indicate that stronger secondary structures tend to occur in longer 3′-UTRs because longer sequences generally have a higher chance of forming stable structures and that these structures, if they really did occur in the samples, did not account for the 3′-UTR effect observed in the data.

Next, we asked whether the effect of 3′-UTR length depends on the stop codon and nt +4 identities. We calculated Spearman's correlation coefficient (ρ) of readthrough efficiency vs. 3′-UTR length for each mRNA group based on their stop codon and nt +4 identities (Supplementary Fig. 5b). We found that the correlation is as high as 0.39 for mRNAs with specific stop codon context, such as UAAG in gentamicin-treated cells and UAGU in tobramycin-treated cells, while

the correlation is close to zero in some other cases. These results demonstrate that the 3′-UTR length effect (and potentially PABP's involvement) may only matter or be observable in some cases, depending on stop codon, nt +4, and aminoglycoside treatment. Although these results are not direct evidence of PABP's involvement in termination, they may still help explain why PABP's role in termination has been elusive, as PABP's role or the extent of its role may be specific to a system or to an mRNA's context.

### Random forest model can accurately predict outcomes of PTC readthrough measured by Dual-Luciferase assay in G418-treated cells

Nonsense mutations introduce PTCs in the middle of mRNA coding regions, resulting in the production of nonfunctional, truncated proteins and severe diseases[31]. Determining the readthrough efficiency of PTCs is crucial in understanding disease phenotypes and designing therapeutic approaches to these diseases. We tested whether the random forest models trained on endogenous mRNA features and transcriptome-wide readthrough efficiency measured by ribosome profiling can accurately predict the readthrough efficiency of 15 PTCs derived from *CFTR* nonsense alleles that are found in cystic fibrosis patients.

To experimentally measure PTC readthrough, we created a dual-luciferase (Dual-Luc) reporter construct for each PTC allele by fusing a DNA fragment containing the PTC and 9 nucleotides (3 codons) flanking the 5′ and 3′ sides of the PTC in-between the *Renilla* and firefly luciferase genes, where the upstream gene (*Renilla*) lacks a stop codon and the downstream gene (firefly) lacks a start codon (Supplementary Fig. 6a). The construct was then transfected into HEK293 cells, treated or not treated with 0.1 mg/mL G418, and expression of the two reporter genes was quantified after 24 h. Readthrough efficiency in 4−7 replicate experiments per allele was determined as the percentage of firefly luciferase normalized to the signal for *Renilla* luciferase (see Supplementary Data 1 for the raw data and calculations for all transformants). To ensure that these enzymatic assays were attributable to stop codon readthrough and not to spurious translation events, e.g., internal translation initiation, that would result in firefly signal, HEK293 cells transformed with the same constructs and treated in the same manner with G418 were subjected to western blotting analyses using anti-*Renilla* and anti-firefly antibodies (Supplementary Fig. 6b, c). All but two alleles, S434X UGA and UAA, of the 15 alleles tested have no other products that would contribute to the luciferase signals (see Materials and Methods for details), so we proceeded with 13 alleles in further analyses.

To predict readthrough, we recorded mRNA features exactly as they appeared in the construct for each PTC allele and used the trained random forest models (one trained on data from untreated cells and another on data from cells treated with 0.5 mg/mL G418) to determine readthrough efficiency values based on the features. The predicted readthrough efficiency is positively correlated with average readthrough efficiency measured in both treatment conditions, with Spearman's rank correlation coefficient ($\rho$) of 0.27 in untreated and 0.7 in the G418-treated condition (Fig. 4a). Excluding the apparent outliers, W882X and W216X, improves the correlation to 0.88 in the G418-treated condition. Predictions from the untreated model performed worse because the model trained on untreated data had poorer performance, or higher NMRSE, than the model trained on G418-treated data (Supplementary Fig. 1a). Since the same mRNA features were used to train both models, the poorer performance is likely due to the lower number of mRNAs with detectable readthrough for model training (Supplementary Fig. 3a) as well as poorer accuracy of readthrough efficiency values derived from ribosome profiling data of untreated cells. Unlike the G418 treatment where readthrough was induced, evidenced by ribosome footprints in the 3′-UTR predominantly in the same reading frame as in the CDS region, ribosome footprints in the

3′-UTR from untreated cells showed no preference in reading frame[17]. Thus, the noise from non-readthrough ribosome footprints was proportionally higher in untreated cells than in G418-treated cells, resulting in poorer predictive ability. Thus, it is unsurprising that when we compared changes in readthrough efficiency upon G418 treatment (G418-treated/Untreated) between predicted and measured values, the correlation was weak (Fig. 4b).

### Three codons flanking each side of a PTC comprise a sufficient context to experimentally measure readthrough efficiency

The random forest models were trained on 75 mRNA features that encompass identities of nucleotides inside and outside the terminating ribosome, properties of amino acids in the ribosome's exit tunnel, and 3′-UTR length (Fig. 1 and Supplementary Fig. 7a, Full). However, the PTC contexts used in the Dual-Luc assays are more limited, only covering 21 mRNA features from the original *CFTR* mRNA sequence (Supplementary Fig. 7a, dark green boxes) and partially affecting 10 nascent peptide tunnel features (Supplementary Fig. 7a, light green boxes), meaning that 44 of the 75 mRNA features used in the prediction were those of the reporter mRNAs that are constant across different alleles. For broader application of the model, predicting readthrough with native sequence features is desirable. Comparing predictions derived from native *CFTR* sequences to alleles with reporter sequences revealed very strong correlations, 0.98 for the G418-treated model (Fig. 4c), suggesting that the 44 mRNA features of the original *CFTR* mRNA that were not copied to the assay had minimal contributions in the prediction. This is expected since the context used in the assays encompasses all the important features except one, the 3′-UTR length (Supplementary Fig. 7a). Additionally, predicted readthrough derived from native *CFTR* sequences still reflected the Dual-Luc assay measurements quite well in the G418-treated condition (Fig. 4d), but not in the untreated condition or in response to G418 treatment (Fig. 4e). The reduction in correlation compared to the analysis derived from reporter sequences was minimal in the G418-treated condition (Fig. 4d vs. 4a), possibly due to the 3′-UTR length feature that varied between PTC alleles in the native sequence but was kept constant in the assay. These results indicate that the PTC context used in the assays may be sufficient to measure readthrough efficiency.

To further ensure that mRNA features outside of the context used in Dual-Luc PTC alleles indeed did not contribute to readthrough efficiency determination, we created additional random forest models lacking those mRNA features and compared the prediction errors to those from the full models for the G418-treated condition (Supplementary Fig. 7). The "Reduced" model contains the combination of features used in the assay and the important features, minus the exit tunnel features (Supplementary Fig. 7a Reduced). Encouragingly, the Reduced model performed as well as the Full model in both regression and classification approaches, as evidenced by unchanged NRMSE and AUROC values (Supplementary Fig. 7b, c, Full vs. Reduced). As controls, we created three "Mock" models that had the same number of features as the Reduced model but only included features with low importance scores (Supplementary Fig. 7a, Mock1−3). In contrast to the Reduced model, Mock models performed significantly worse than the Full model in both approaches, as evidenced by increases in NRMSE and decreases in AUROC values (Supplementary Fig. 7b, c, Full vs. Mock). Because the Dual-Luc assay did not take into account the 3′-UTR length, which is identified as an important predictor of readthrough efficiency in random forest models (Fig. 1), we next asked whether excluding 3′-UTR length from the Reduced model significantly changed model performance (Supplementary Fig. 7a, Assay). This model performed slightly worse than the Full model, but not significantly (Supplementary Fig. 7b, c, Full vs. Assay). These results show that stop codon context alone is sufficient to predict readthrough efficiency.

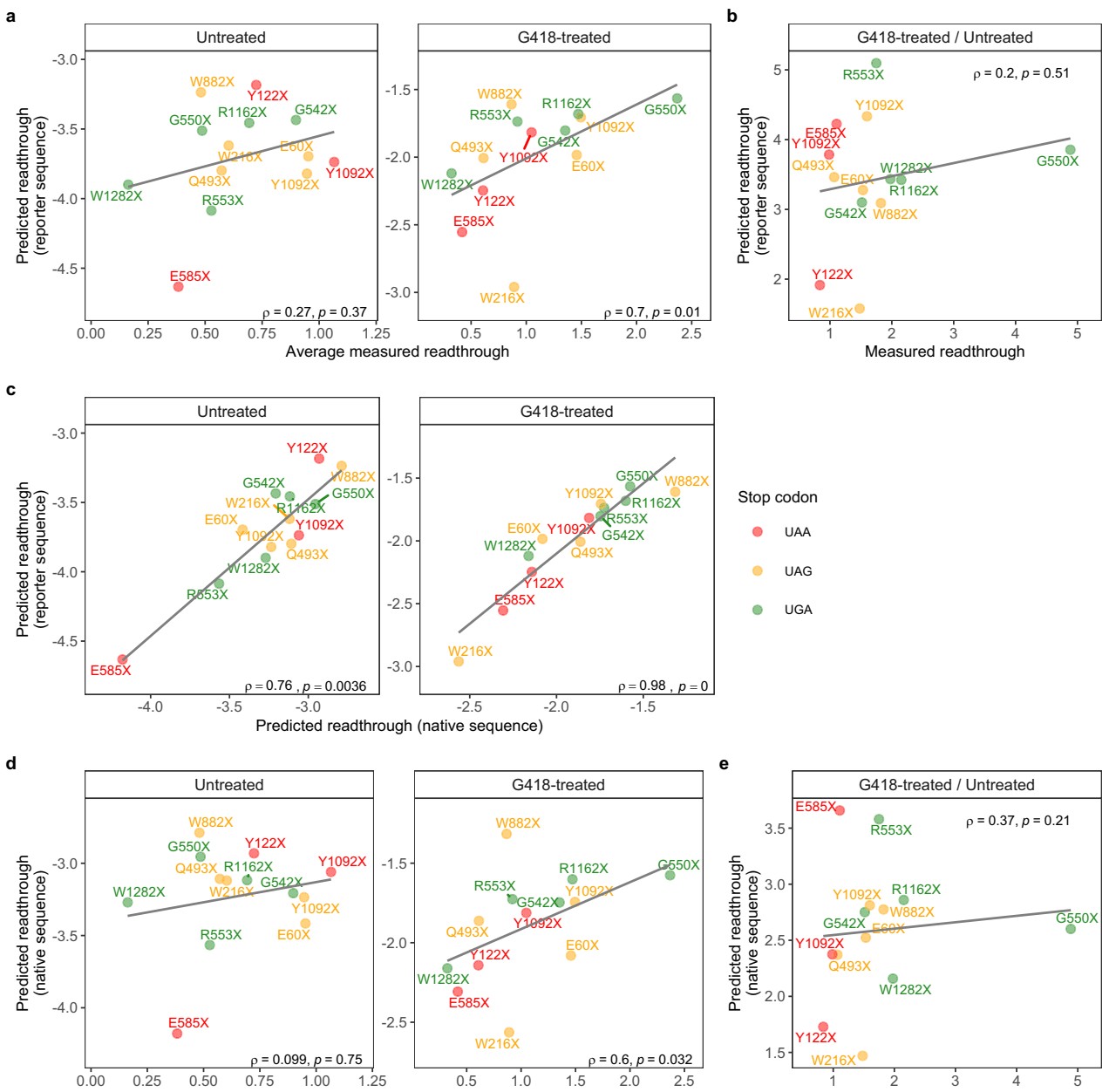

**Fig. 4 | Random forest model can accurately predict the readthrough of *CFTR* PTC alleles in G418-treated cells. a** Readthrough measured by Dual-Luc assay (average of 4–7 replicates) vs. readthrough predicted by random forest model using *CFTR* PTC alleles in Dual-Luc reporter's sequence. **b** Response to G418 treatment is defined as fold-change of readthrough in G418-treated to untreated condition from (**a**), measured vs. predicted. **c** Comparison of two readthrough prediction schemes: predicted using *CFTR* PTC allele in reporter's sequence or predicted using *CFTR* PTC allele's native sequence. **d** As in (**a**), but readthrough was predicted using *CFTR* PTC allele's native sequence. **e** As in (**b**), but readthrough was predicted using *CFTR* PTC allele's native sequence. For all panels, the two-tailed Spearman's correlation coefficient ($\rho$) and the associated *p*-value is reported. Source data are provided as a Source Data file and Supplementary Data 1.

Together, our findings not only demonstrated a broader application of the model in predicting readthrough efficiency but also indicated that three codons flanking each side of a PTC is a sufficient context to experimentally measure readthrough efficiency.

**Stop codon context in *CFTR* PTC alleles**

Next, we asked what contexts existed in PTC alleles from highest to lowest readthrough measured in the G418-treated condition and whether they were consistent with the results derived from transcriptome-wide ribosome profiling data of G418-treated cells (Supplementary Fig. 8). Notably, the allele with the highest readthrough, G550X, has a UGA stop codon and readthrough-promoting

context for nt −1 and +9 (Supplementary Fig. 8a). The highest readthrough among UAG and UAA alleles, Y1092X-UAG and -UAA, have readthrough-promoting C at the nt +4 and A at the nt +9 positions (Supplementary Fig. 8b). Comparing to its UAA counterparts with identical context, Y1092X-UAG has higher readthrough than Y1092X-UAA, as expected (Supplementary Fig. 8a). These relative comparisons are also reflected in predicted readthrough of these alleles by the Dual-Luc assay (Fig. 4a).

Among the three UAA alleles, E585X unsurprisingly has the lowest readthrough because it has the most readthrough-inhibiting context UAAA (Supplementary Fig. 8b). However, two of the UAA alleles actually have very similar immediate context, UAACU, but their measured read through varied, suggesting the roles of

other nucleotide positions such as −1, and +9 in mediating readthrough efficiency (Supplementary Fig. 8b). For UAG alleles, the trend is more difficult to discern, but if the two outliers (W882X and W216X) where the predicted and measured readthrough do not quite agree (Fig. 4a) are excluded from consideration, the rank order of Y1092X, E60X, and Q493X suggests the role of nt +4 and +9 (Supplementary Fig. 8b).

Although W1282X being the lowest UGA readthrough allele is not surprising considering its readthrough-inhibiting context, it is surprising that it actually has even lower readthrough than the lowest UAA readthrough allele, E585X, as evidenced by the discrepancy between the measured and predicted readthrough ranking of the two alleles (Fig. 4a). This discrepancy, among other outliers, suggested that there exist other features that influence readthrough efficiency that were not included in the model (see Discussion). Nevertheless, the overall high correlation of predictions using the current model and assay measurements (Fig. 4), as well as the generally expected preference for readthrough-promoting or -inhibiting contexts in highest and lowest readthrough alleles (Supplementary Fig. 8), demonstrate value in the application of machine learning in readthrough efficiency prediction and validate the use of limited context in measuring readthrough experimentally.

## Mutations that result in UAA PTCs do not respond well to G418 treatment

One of the major questions in nonsense suppression therapeutics is how responsive a PTC is to drug treatment. Because the model did not give accurate predictions in the untreated condition (Fig. 4b, e) to use as baseline readthrough efficiency, we investigated response to G418 treatment using Dual-Luc results (Fig. 5). It appears that UAA alleles generally did not respond well to G418 treatment (Fig. 5a). G550X showed the best response to G418 treatment, showing the highest increase in readthrough, although it was impossible to determine the reason without further experiments or more data (Fig. 5a). However, we observed a trend that UGA alleles as a group had the highest, followed by UAG, and UAA alleles the lowest increase in readthrough upon G418 treatment (Fig. 5a, b). Thus, in addition to being readthrough-inhibiting in general, UAA also seems to inhibit G418-mediated induction of readthrough.

## Discussion

### Major *cis*-acting elements modulating transcriptome-wide translation termination and readthrough efficiency are conserved

Here, we analyzed published readthrough efficiency data derived from ribosome profiling of HEK293T cells[17] using analysis approaches we had previously developed with yeast data[16]. The unbiased random forest approaches identified the same elements modulating readthrough efficiency in both human and yeast cells, which include the stop codon, nt +4, 3'-UTR length, and P-site amino acids (Fig. 1). Among these elements, the stop codon and nt +4 (as well as readthrough promoting nucleotides at +5 and +9), which all have high (or relatively high) feature importance scores, influence readthrough efficiency in the same manner between the two organisms (Figs. 2a and 6)[16]. These results are in line with previous knowledge of *cis*-acting elements that modulate termination, extending that knowledge to a transcriptome-wide level and reinforcing the notion that the mechanism of termination is conserved in eukaryotes.

The exact mechanism of nt +4 and +5 in influencing termination has been demonstrated in cryo-electron microscopy (cryo-EM) studies, showing that mRNA compaction during termination allows the +4 base to enter the ribosomal A-site, making a +4 base stack with $G_{626}$ of 18 S rRNA[14,15,32] and +5 base stack with $C_{1698}$ of 18 S rRNA that protrudes into the mRNA channel[15]. These stacking interactions are more stable with purines (A and G), which explains why stop codons followed by a purine, especially G, at nt +4 result in lower readthrough efficiency. However, A at nt +5 is associated with higher readthrough efficiency in our analysis and many previous studies[4,6,9,22]. Given that there are studies that did find A at nt +5 to be readthrough-inhibiting[10,25], the discrepancy between what the cryo-EM implies and some results may be related to differences in reporter sequences, indicating complex roles of other nucleotides in the overall structure of the terminating ribosome. Although the exact mechanism of further downstream nucleotides, including nt +9, is still unclear, it is possible that nt +9 also interacts with an rRNA base or an amino acid residue(s) of a ribosomal protein because it appears to reside in the ribosome's mRNA channel. The latter conclusion follows from their protection from RNaseI digestion in ribosome profiling experiments as well as inferences of structural studies[8,20,33–35]. A recent study using a reconstituted

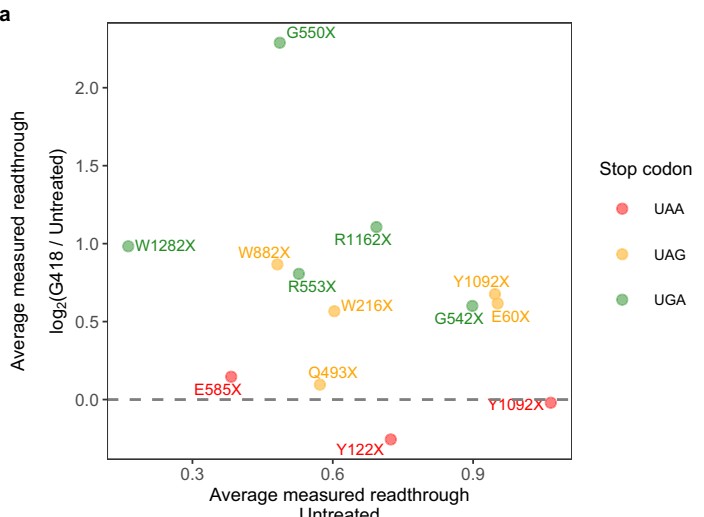

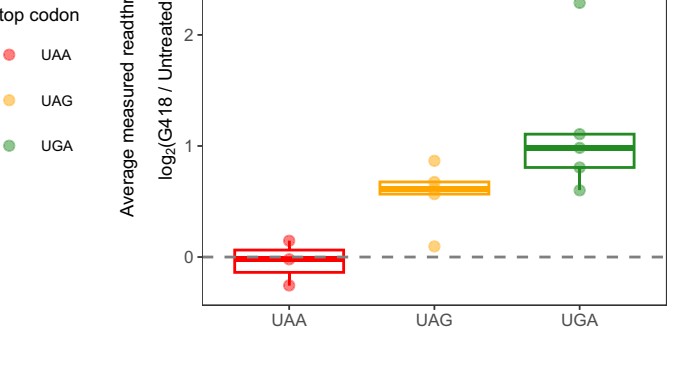

**Fig. 5 | UAA PTC alleles do not respond well to G418-treatment.** Response to G418 treatment is defined as $\log_2$ fold-change of readthrough for each replicate of G418-treated over the untreated condition and then averaged (4–7 replicates). **a** Average response to G418 treatment vs. average basal readthrough level in an untreated condition. **b** Average response to G418 treatment vs. stop codon identity.

Box-plot center line, median; lower and upper hinges, first and third quartiles (the 25th and 75th percentile); whiskers, 1.5× interquartile range; points, actual data. Two-tailed Student's *t*-test with the Benjamini–Hochberg method for multiple testing correction was used to perform the pairwise comparison. Source data are provided as a Source Data file and Supplementary Data 1.

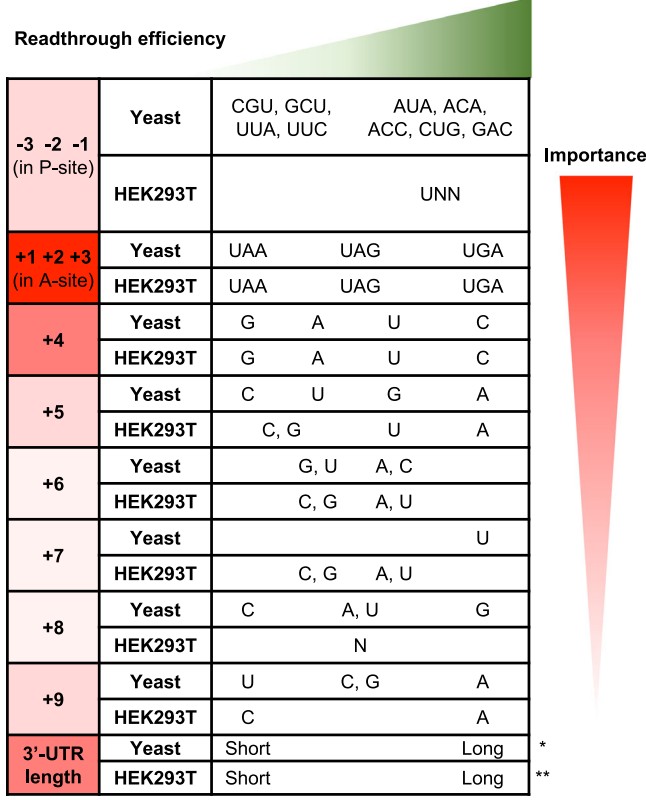

**Fig. 6 | *Cis*-acting elements modulating readthrough efficiency are mostly conserved between yeast and humans.** Comparison of results analyzed in a similar manner with data from yeast W303[16] and human HEK293T cells[17]. mRNA features are shaded based on relative importance in readthrough efficiency prediction (red shading). The details of how each feature inhibits or promotes readthrough are ordered from left to right (green shading).

translation system further demonstrated that readthrough efficiency varied in a context-dependent manner in the absence of eRF1, while termination efficiency did not differ between standard and weak context, leading to a proposition that downstream nucleotides affect the rate of near-cognate tRNA incorporation (rather than eRF1's stop codon recognition or peptide release activity) through the involvement of ribosomes and other factors usually engaged in translation elongation[36].

One difference between the data from HEK293T cells and yeast is the relationship between readthrough efficiency and 3'-UTR length in wild-type conditions. Under circumstances where termination is rendered less efficient, either by release factor deactivation in yeast or by aminoglycoside treatment in HEK293T cells, readthrough efficiency increases with 3'-UTR length, implying that the closer proximity of PABP to the stop codon enhances termination efficiency when the normal termination process is compromised. Under normal circumstances where termination is not hindered, readthrough still increases with 3'-UTR length in untreated HEK293T cells (Figs 3 and 6), but decreases with 3'-UTR length in wild-type yeast cells[16]. This discrepancy may be related to the fact that yeast mRNA 3'-UTRs are naturally much shorter than those of human cells (Supplementary Fig. 4b), limiting possible additional 3'-UTR interactions and regulation. This reduced binding opportunity for additional regulatory proteins to the 3'-UTR and increased proximity of poly(A) tail (and poly(A) tail-associated proteins) to the stop codon may suggest that these regulatory factors may have either stronger or weaker influence on 3'-UTR function under normal circumstances in yeast compared to human cells.

## tRNA abundance and/or properties may influence the efficiency of translation termination

Contrary to other mRNA contexts where the trends are similar between human and yeast data, the identity of P-site codons that are associated with high or low readthrough efficiency mostly differ between human and yeast cells (Figs. 2d and 6). Except for the GCU codon that is associated with low readthrough almost uniformly across samples in both species, no other codons in human cells showed patterns consistent with those found to be significant in yeast cells. Because the nucleotide and amino acid properties do not vary between species, the differences in results could be attributable to structural variation between yeast and human ribosomes or the difference in tRNA pools and properties. For the latter scenario, even though the mechanism underlying the influence of P-site position in readthrough efficiency may be conserved, the variation in tRNA abundance and modifications between species can result in differences in the identity of the codon identified in our P-site codon analyses. Although P-site tRNA was claimed to have no effect on readthrough levels, this conclusion was derived from observed changes in readthrough levels when the third wobble base of the codon was mutated while keeping the same decoding tRNA for both the original and the mutated codons unchanged[37]. Additional experiments that can control for nucleotide and amino acid identities while varying the tRNA properties will be needed to pinpoint the underlying mechanism behind the P-site codon's effect on readthrough efficiency.

In addition to the P-site, tRNA abundance and properties likely also matter in stop codon decoding in the A-site. Termination vs. readthrough is a competition between eRF1 and a near-cognate tRNA in A-site binding. Several studies in yeast and human cells have shown that specific near-cognate tRNAs are more readthrough-inducing than others[38–40], which are influenced by their relative concentration in the cells[39,41], modifications of their anticodon loop that interacts with the stop codon[40,42], and properties of their anticodon stem that interacts with the ribosome[43,44]. These features are not included in the random forest models, and the missing information may be one of the reasons for the existence of outliers in the comparison of predicted and measured readthrough efficiency (Fig. 4). In particular, we noticed that the predictions are the least accurate for UAG PTC alleles; Spearman's correlation coefficients for the data in Fig. 4a, G418-treated panel, for each stop codon are 0.9 for UGA, 0.2 for UAG, and 1 for UAA. Although the same amino acids are usually inserted at UAA and UAG stop codons, the frequencies differ, and the near-cognate tRNAs that decode them are not always the same[38–40,45,46]. Perhaps in HEK293T cells, the concentration of some tRNAs that decode UAG is much higher or lower than average, making it a rate-determining or rate-limiting step in the eRF1 vs. tRNA competition and overpowering noticeable influences of other mRNA features. To add to the complexity of this process, stop codon context not only influences readthrough efficiency, but also dictates which tRNA and thus amino acid to be inserted at a PTC[46,47]. An additional consideration is the stop codon and near-cognate tRNA base-pairing kinetics. Previous studies in both yeast and human cells demonstrated that the 2nd base (middle) is almost always cognate while mismatches occur at either the 1st or 3rd base[38,39,45,46]. This means that UAA and UGA have the same nucleotides for the 3rd base mismatch, A, while UAG has G. Again, the different wobble base-pairing kinetics with G as well as properties of a different tRNA decoding UAG, may be rate-determining or rate-limiting for readthrough, resulting in the predictions based on other mRNA features alone being less accurate.

Overall, our results suggest that tRNA abundance and/or properties may influence readthrough efficiency. Including precise information regarding tRNAs in the model will likely improve its predictive ability.

### Potential applications of machine learning models in read-through prediction

Apart from a few outliers, we showed that random forest models trained on readthrough data of endogenous mRNAs derived from ribosome profiling experiments can accurately predict the readthrough of *CFTR* PTCs measured by a Dual-Luciferase assay, particularly in G418-treated cells (Fig. 4). The significance of this result is three-fold.

First, this result indicates that readthrough efficiency at NTCs and PTCs share the same rules for *cis*-regulatory elements. Because nonsense mutations happen randomly and are not subject to the same evolutionary pressure as that of NTCs, there has been a concern that information obtained from studying NTC readthroughs may not be relevant to studying disease-related nonsense mutations. However, we showed that despite biases in certain contexts appearing more often or occurring less frequently in the transcriptome (Fig. 2c), the endogenous mRNA NTC contexts are diverse enough to allow accurate prediction of the readthrough of PTCs (Fig. 4).

Second, this result validates the use of a small fragment of native PTC context in studying readthrough. Studies of readthrough usually involve only a fragment of native sequence near the stop codon, which varies between studies. It was unclear whether this small fragment would be sufficient to mimic the readthrough of PTCs with the whole-gene native sequence that exists in patients. Our trained model includes larger native sequence contexts than are used in the Dual-Luciferase assay, but the prediction and measured readthrough efficiency are still consistent with each other. Hence, we provided evidence that minimal context that contains the most influential mRNA features is sufficient to experimentally measure readthrough.

Lastly, our results demonstrate that a machine learning model trained on big data can predict an assay outcome. Studying readthrough of nonsense mutations in disease patients has mostly required difficult-to-estimate, indirect measurements of readthrough[48–51]. The Dual-Luciferase assay, although commercialized and streamlined for ease of use, can still be tedious if there are many mutations and contexts to test. Thus, the high predictive ability of a machine learning model may help with future readthrough experimental design, saving time, labor, and resources. Although basal readthrough prediction is not as accurate, making studying fold-change in drug treatment more difficult to predict, it might not matter because the most important information for therapeutic purposes is readthrough level after drug treatment, which we showed to be accurate. With additional consideration for how the inserted amino acid affects the full-length protein function, the application of readthrough prediction may also extend to aiding clinical trial design for nonsense suppression therapies, predicting whether a patient will respond well to a readthrough drug given the nonsense mutation's sequence context.

## Methods

### Data acquisition

Readthrough efficiency data for HEK293T cells treated with different aminoglycosides were downloaded from Wangen and Green, *eLife*[17], Fig. 2—Source Data 1[17] and log$_2$-transformed for all analyses.

### mRNA sequence features

Sequences of spliced mRNAs were downloaded from Ensembl using R package biomaRt[52] according to Ensembl Transcript ID provided in Wangen and Green, *eLife*[17] Fig. 2—source data 1. The final HEK293T mRNA isoforms selection was described in detail in the original publication[17], and mRNA region lengths provided in Fig. 2—source data 1 were adjusted accordingly to reflect actual lengths as well as to consider the stop codon as part of the 3'-UTR. For yeast, the longest UTR entry was chosen for mRNAs with multiple annotations across studies[53–57] deposited in the YeastMine database (as of July 3, 2017), and mRNA region lengths were adjusted accordingly to consider the stop

codon as part of the 3'-UTR. Other HEK293T mRNA sequence features were defined in the same manner we previously utilized for yeast[16].

mRNA secondary structure in the 3'-UTR sequence was predicted using RNALfold (default parameter, -L 150) of ViennaRNA package version 2.6.3[30]. In the case that no structure was predicted for a given 3'-UTR, the minimum free energy (MFE) of zero was assigned. In the case that multiple structures were predicted for a given 3'-UTR, the MFE of the most stable structure for that 3'-UTR (lowest MFE) was used for analysis.

### Random forest models and statistical analyses

Analyses were performed in the R programming environment version 3.6 on the UMass high-performance computing cluster (HPCC) and version 4.2 on macOS Monterey 12.6.3. The following R packages were used with R version 3.6: caret (v 6.0-86) and randomForest (v 4.6-14). The following R packages were used with R version 4.2: readxl (v 1.4.0), data.table (v 1.14.2), dplyr (v 1.0.8), reshape2 (v 1.4.4), biomaRt (v 2.52.0), randomForest (v 4.7-1), caret (v 6.0-92), Biostrings (v 2.64.0), seqinr (v 4.2-8), rstatix (v 0.7.0), ggplot2 (v 3.4.0), ggpubr (v 0.4.0), ggh4x (v 0.2.3), ggrepel (v 0.9.1), scales (v 1.2.1), patchwork (v 1.1.1), and Cairo (v 1.5-15).

For each sample, mRNAs with too little read coverage and undetectable readthrough, where RPKM of CDS < 5 and RPKM of the extension region (3'-UTR region between the canonical stop codon and next in-frame downstream stop codon) <0.5, were discarded from further analyses. The remaining data was used to create a random forest regression model and for comparative analyses. The top 15% and bottom 15% of the remaining data ranked by readthrough efficiency were assigned as "high" and "low" readthrough mRNAs, respectively, and used to create a random forest classification model.

Random forest models were trained to predict readthrough efficiency (regression) or readthrough groups (classification) with 100 trees and 5-fold cross-validation to optimize the number of features allowed for splitting at each node (mtry hyperparameter). For additional models in Supplementary Fig. 7, the same parameters were applied, except mtry was kept as default. Performance metrics extracted from each model are root mean squared error (RMSE) normalized to the range of *Y* variable (readthrough efficiency) for the regression model and area under the receiver operating characteristic (AUROC) for the classification model. Feature importance metrics, which indicate the predictive ability of each mRNA feature, extracted from the final model are % increase in mean squared error (%IncMSE) for the regression model and Mean Decrease Accuracy (MDA) for the classification model.

Comparative analyses between the median readthrough efficiency of a group of mRNAs (defined by the identity of stop codon, nucleotide, or codon triplet) and the median readthrough efficiency of all mRNAs in a sample ("sample median") were performed by two-tailed Wilcoxon's rank sum test with Benjamini-Hochberg multiple testing correction.

### Generation of dual-luciferase h*CFTR* PTC reporters

Sense/antisense oligonucleotides containing h*CFTR* PTC contexts (PTC plus 3 codons of upstream and downstream h*CFTR* context) were annealed and cloned into the AscI/Sbf1 sites of dual-luc AscI G542X 11 codon SbfI/pcDNA3.1 Zeo+ (pDB1497). The annealed oligonucleotides were inserted between upstream *Renilla* and downstream firefly luciferase genes. The firefly activity can only be detected when a readthrough of the PTC occurs. A total of 15 dual-luciferase reporters containing different h*CFTR* PTC contexts (6 UGAs, 5 UAGs, and 4 UAAs) were generated. The plasmids and oligonucleotide sequence information are provided in Supplementary Data 2.

For readthrough prediction of h*CFTR* PTC reporters, the sequences of luciferase genes were used to determine mRNA features outside of the PTC contexts. The 3'-UTR length is defined as the

number of nucleotides from PTC to the bGH poly(A) signal in the reporter.

## Dual-luciferase reporter assay to measure PTC readthrough in HEK293 cells

The dual-luciferase reporter constructs were transiently transfected into HEK293 cells (CLS Cat# 300192/p777_HEK293, RRID:CVCL_0045) to test readthrough efficiency. HEK293 cells were seeded into 96-well plates at $2 \times 10^4$ cells/well. Twenty-four hours after seeding, the cells were transfected with 0.1 μg DNA/well using Lipofectamine LTX reagents (ThermoFisher Cat# 15338500). Three hours after transfection, the cells were treated with 100 μg/ml G418. After 24 h of treatment, dual luciferase assays were performed using a Dual-Luciferase Reporter Assay System (Promega Cat# E1960) with a Glomax Discover Microplate Reader (Promega) to measure both *Renilla* and firefly activities. The firefly activity normalized to the *Renilla* activity was calculated as:

$$\text{Readthrough } level = \frac{\text{firefly } units}{\text{Renilla } units} \times 100 \qquad (1)$$

Three parameters were reported from this assay: (1) the basal readthrough in untreated cells; (2) the readthrough induced by 100 μg/ml G418; and (3) the fold-increase in readthrough when comparing G418-treated and untreated samples. Each experiment included treated and untreated samples assayed in quadruplicate wells in a 96-well plate. Outliers within each set of replicate wells were defined as replicates that made standard deviation/average (SD/AVE) % >25%. They were excluded from AVE and SD calculations and indicated in red font in Supplementary Data 1. Four to seven independent experiments were performed for each h*CFTR* PTC dual-luc reporter.

## Western blotting confirmation of dual-luciferase enzymatic assays

To confirm that the dual-luciferase constructs were expressed as expected, western blotting with antibodies targeting *Renilla* and firefly luciferases was employed to identify the respective polypeptide products. HEK293 cells were transiently transfected with dual-luciferase reporters and treated with G418 as above, then lysed with M-PER Mammalian Protein Extraction Reagent (Thermo Scientific Cat. #78501), including protease inhibitors. Aliquots (20 μg protein) of each sample were fractionated by SDS-PAGE and subjected to western blotting using the following primary and secondary antibodies: mouse anti-tubulin (DSHB E7; 1:1000 dilution), rabbit anti-Rluc (Invitrogen PA5-32210; 1:500 dilution), rabbit anti-Fluc (Invitrogen PA5-32209; 1:2000 dilution), LI-COR IRDye® 680RD Goat anti-Rabbit IgG Secondary Antibody (LI-COR Cat. # 926-68071, 1:20,000 dilution), and LI-COR IRDye® 800CW Goat anti-Mouse IgG Secondary Antibody (LI-COR Cat. # 926-32210, 1:20,000 dilution). Termination at the PTC of the dual-luciferase construct should yield a 37 kDa *Renilla* luciferase polypeptide, whereas readthrough of the PTC should yield a 100 kDa fusion protein consisting of a fusion of *Renilla* and firefly luciferases. *Renilla* antibody generally detected a strong 37 kDa band and a much weaker 100 kDa band; the firefly antibody detected the full-length 100 kDa band more efficiently than the *Renilla* antibody.

## Reporting summary

Further information on research design is available in the Nature Portfolio Reporting Summary linked to this article.

## Data availability

Source data are provided with this paper. Raw luciferase signals and the firefly/Renilla ratios are provided in Supplementary Data 1. All processed data and each figure's source data are available in the Source Data File and without restriction at https://github.com/Jacobson-Lab/ AG_readthrough (https://doi.org/10.5281/zenodo.10698037)[58]. Databases employed in this study include YeastMine (https://yeastmine.yeastgenome.org/yeastmine/begin.do) and Ensembl (https://useast.ensembl.org/index.html). Source data are provided in this paper.

## Code availability

Scripts used to acquire, analyze, and visualize data are available without restriction at https://github.com/Jacobson-Lab/AG_readthrough (https://doi.org/10.5281/zenodo.10698037)[58].

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

## Acknowledgements

This work was supported by grants from the U.S. National Institutes of Health 1R35GM122468 (to A.J.), the Cystic Fibrosis Foundation CFFKOROST20GO (to Andrei Korostelev and A.J.), and the Cystic Fibrosis Foundation BEDWEL19G0 and Southern Research subaward 858345 (to D.B.). We thank members of the Jacobson lab for comments on the manuscript.

## Author contributions

K.M. and A.J. conceived and designed the analysis; K.M. carried out the bioinformatics analysis; L.F., M.D., K.T., and K.K. conceived and designed

the dual-luciferase *hCFTR* readthrough experiments; L.F., M.D., and K.T. carried out the dual-luciferase *hCFTR* experiments; K.M., L.F., M.D., K.T., K.K., D.B., and A.J. analyzed the data; K.M. and A.J. wrote the paper; A.J. and D.B. obtained funding for the study.

## Competing interests

A.J. is co-founder, director, and consultant for PTC Therapeutics Inc. D.B., K.M., L.F., M.D., K.T., and K.K. declare no competing interests.
