## [Peer Review File · Nature Communications]

Extended stop codon context predicts nonsense codon readthrough efficiency in human cellsREVIEWER COMMENTS

Reviewer #1 (Remarks to the Author):

Mangkalaphiban et al used a machine learning approach to re-analyze ribosome profiling data generated in G418-treated HEK293T cells by Wangen and Green in 2020. Using this approach they re-confirm that the stop codon identity and +4 nt are strong determinants of readthrough. The C-terminal amino acid, certain other 3' positions and 3'UTR length are also shown to influence readthrough efficiency. For me, the most important part of this study is the ability to predict relative readthrough potential on PTCs in cells treated with potential therapeutics (small molecules or suppressor tRNAs) from the readthrough scores of NTCs determined by ribosome profiling. Although NTCs are under positive selection for efficient termination, I agree with the authors that there should be enough context diversity around NTCs to train their predictive models, especially in cells treated with potential therapeutics. Although I don't think this is required for publication, the authors may be able to improve their model by additionally training it on read through uORF stop codons which are likely under less evolutionary selection for efficient termination (in theory they could be viewed as PTCs and there are very few mammalian mRNAs without translation in their 5'UTRs). Overall, I am positive about publication in Nature Communications although there is one technical issue which should be addressed.

The authors approach to determining readthrough efficiencies by dual luciferase assays is highly unusual. Normally readthrough efficiency is determined relative to a sense codon control (in-frame control equivalent to 100% readthrough). The main reason for this is that the absolute luciferase values for Renilla and firefly are rarely (if ever) the same. The readthrough efficiencies calculated in this manuscript rely on the absolute luciferase values of Renilla and firefly being identical (if expressed from an in-frame control). This means that the readthrough efficiency calculations are unreliable and can only be compared relative to each other (which is fine for this study). However, even comparing relative to each other may be problematic here as the authors do not provide the absolute luciferase values (or at least I couldn't find them). For these readthrough values to be comparable relative with each other the absolute Renilla values should be similar between different reporter constructs. Please include the absolute luciferase values in the extended data.

Discussion - reenforcing

Reviewer #2 (Remarks to the Author):

In this manuscript, Mangkalaphiban et al. adopted a machine learning approach, developed previously by them for yeast studies, to analyze readthrough efficiency data from published HEK293T ribosome profiling experiments. Evidence for the conservation of identities of stop codon, its close context, and partially also 3'-UTR length is presented. They also show that models trained on data from G418-treated cells predict readthrough of premature termination codons arising from CFTR nonsense alleles.

Overall, using the computer-based approach, the authors confirmed most of the previous findings and reassured the community that "the endogenous mRNA NTC contexts are diverse enough to allow accurate prediction of readthrough of PTCs", which I find important. In addition, they also documented that "the use of a small fragment of native PTC context" is sufficient for studying a SC-RT potential of a given naturally occurring PTC.

Specific comments, suggestions (in the order of their appearance in the text):

- Abstract: "suggestion a tRNA role in readthrough regulation"; did you mean a P-site tRNA? Please specify.

- Page 4; "As expected, the negative control ..."; this sentence is confusing.
- The fact that the untreated model had a low number of mRNAs with detectable SC-RT is, in my opinion, a major, unfortunate setback of this study. For example, even stop codons and nt +4 show low score in Fig. 1 (in contrast to 3' UTR, which is a bit puzzling for me), and predictions from the untreated model do not look very convincing (Fig. 4).
- Page 5, very top, „general trends“. I think it would interesting to pick all genes that are known to allow SC-RT and show whether their stop codon contexts comply or not with these trends.
- Figure 2D; I think the striking case to comment on is ACG, which worked well even in untreated but for some reason was overlooked. None of the other three Thr codons showed such specificity.
- As for the 3' UTR effect, have the authors considered analyzing the role of prospective sec. structures immediately following stops (mentioned in the intro)?
- Dual Luc measurements. They are mentioned in the article, but they are not actually shown (just raw measurements as appended xls files), nor is there any reference to them in the text (where to find them). This should be corrected.
- E585X seems to be even a bigger outlier than W822X, at least in untreated. Why was it ignored?
- Does it really make sense to consider the impact of the 3' UTR length when it comes to PTCs (Ext. Data Fig. 4)?
- Discussion; I would be interested to hear the author's thoughts on the mechanistic contribution of +5 and +9 bases (mentioned on page 11) to SC-RT.
- Page 12, top paragraph. Our and O. Namy's labs have also contributed quite extensively to elucidation of the "base-pair kinetics" in yeast and mammalian cells, in other words, to deciphering which nc-tRNAs do and which don't promote SC-RT and why, so I humbly ask the authors to cite at least some of these studies. This is indeed just a comment, not a requirement.

Thank you for giving me the opportunity to review this article. Leos S. Valasek

Reviewer #3 (Remarks to the Author):

In this manuscript, Mangkalaphiban and colleagues apply a machine learning approach to reanalyze ribosome profiling data from HEK293 cells to assess readthrough efficiency. They verified the impact of features such as the termination codon identity and the importance of the fourth nucleotide and suggested a role of long 3' UTRs in enhancing readthrough. They also suggest the adoption of machine learning in readthrough prediction, which is also an interesting application.

1. Results, page 7: "Readthrough efficiency increases with 3' UTR length": The p values of Figure 3 are too low to allow such a bold statement. Moreover, there is thin evidence in mammalian systems that PABPC plays a role in efficient translation termination based on an in vitro translation system under limiting release factor concentrations.
2. Results, page 4: I failed to find in Figure 1 the feature of the long 3' UTRs, and therefore, I cannot assess the corresponding readthrough efficiency. Please clarify which part of the figure describes the effect of the 3' UTR length.

Considering points 1 and 2, I cannot conclude a clear correlation between the 3' UTR length and

readthrough efficiency, which is the only novel feature that this analysis proposes.

3. There are increasing concerns in the translation community about using bicistronic reporter assays to assess translation efficiency, especially based on DNA transfections which can additionally lead to transcription/splicing aberrancies. There is also evidence that test sequences placed between two reporter genes can inadvertently modify the expression or functionality of either reporter. This can result in misleading outcomes, potentially leading to inaccurate findings published in the scientific literature (for instance, <https://doi.org/10.1073/pnas.2122170119>). For this reason, given that Dual luciferase assays are the only additional experimental input in this manuscript (Fig.4), it is suggested at least to perform RNA transfections instead of DNA-based assays.

4. Results, page 5: "Readthrough-promoting stop codon and nt+4 occur together less often than expected": The title, as well as the text of this part of the text, was hard to comprehend. The last paragraph provides an overview of the finding is much more clear than the rest of the text.

- It is suggested to clarify further what the authors mean by the phrase "allow chances for protein adaptation". Some sentences are too long "To statistically determine... stop codons", which hinders comprehension.

- The phrase "For the reference and G-418... over-represented" was not clear to me. What does it mean that their association is truly over-represented?

RESPONSES TO THE COMMENTS FROM REVIEWERS

Reviewer #1 (Remarks to the Author):

Mangkalaphiban et al used a machine learning approach to re-analyze ribosome profiling data generated in G418-treated HEK293T cells by Wangen and Green in 2020. Using this approach they re-confirm that the stop codon identity and +4 nt are strong determinants of readthrough. The C-terminal amino acid, certain other 3' positions and 3'UTR length are also shown to influence readthrough efficiency. For me, the most important part of this study is the ability to predict relative readthrough potential on PTCs in cells treated with potential therapeutics (small molecules or suppressor tRNAs) from the readthrough scores of NTCs determined by ribosome profiling. Although NTCs are under positive selection for efficient termination, I agree with the authors that there should be enough context diversity around NTCs to train their predictive models, especially in cells treated with potential therapeutics. Although I don't think this is required for publication, the authors may be able to improve their model by additionally training it on readthrough uORF stop codons which are likely under less evolutionary selection for efficient termination (in theory they could be viewed as PTCs and there are very few mammalian mRNAs without translation in their 5'UTRs). Overall, I am positive about publication in Nature Communications although there is one technical issue which should be addressed.

Response: We thank the reviewer for assessing our work positively. The reviewer raised a good point about the potential benefit of including uORF readthrough in the model due to uORF stop codons being under less evolutionary selection. However, we found that the relatively low yield of reads from mRNA 5'-UTRs seemed to make accurate determination of readthrough efficiency very difficult. Ribosome profiling libraries, as expected, consist mostly of reads mapped to the coding region (Fig. 1, adjacent; CDS, grey), with readthrough of CDS stop codons making up, at most, only 2.74% of all reads in a G418-treated sample (Fig. 1, 3'-UTR, blue). Assuming that ~5% of reads mapped to 5'-UTRs account for uORF translation and that the uORF readthrough rate is similar or even slightly higher than the CDS readthrough rate, the number of reads for uORF readthrough would be too low for accurate measurement of readthrough efficiency, which is critical in studying the relevant *cis-acting* elements. Thus, we have decided to not include uORFs in our analyses.

Figure 1. Percentage of ribosome profiling reads (y-axis) mapped to each mRNA region (colors) in each sample (x-axis).

The authors approach to determining readthrough efficiencies by dual luciferase assays is highly unusual. Normally readthrough efficiency is determined relative to a sense codon control (in-frame control equivalent to 100% readthrough). The main reason for this is that the absolute luciferase values for Renilla and firefly are rarely (if ever) the same. The readthrough efficiencies calculated in this manuscript rely on the absolute luciferase values of Renilla and firefly being identical (if expressed from an in-frame control). This means that the readthrough efficiency calculations are unreliable and can only be compared relative to each other (which is fine for this study). However, even comparing relative to each other may be problematic here as the authors do not provide the absolute luciferase values (or at least I couldn't find them). For these readthrough values to be comparable relative with each other the absolute Renilla values should be similar between different reporter constructs. Please include the absolute luciferase values in the extended data.

Response: The raw measurements for *Renilla* and firefly luciferase activity, and their calculated ratios, were originally provided in Supplementary Table 2 (now Extended Data Table 1), an updated copy of which is included with our revised manuscript. All raw values used are shown. Only the values indicated in red that were determined to be >2SD from the mean were excluded from the final calculations.

We contend that normalizing to wild-type firefly/*Renilla* values is not necessary and will only introduce additional error (taking a ratio of a ratio, e.g., PTC FF/*Renilla* x 100 / WT FF/*Renilla* x 100). Please note that while there is some variability among the absolute *Renilla* luciferase readings, as one would expect from differences in transfection efficiencies, etc., the normalization of firefly to *Renilla* resolves this variability in the readthrough efficiency calculations. This point is illustrated more extensively in the appended PPT file **Examples of Calculations 1-4-24**.

In addition, please see the response to Reviewer #3 (below) for details of our newly added approach to validating the results of our dual luciferase assays.

Reviewer #2 (Remarks to the Author):

In this manuscript, Mangkalaphiban et al. adopted a machine learning approach, developed previously by them for yeast studies, to analyze readthrough efficiency data from published HEK293T ribosome profiling experiments. Evidence for the conservation of identities of stop codon, its close context, and partially also 3'-UTR length is presented. They also show that models trained on data from G418-treated cells predict readthrough of premature termination codons arising from CFTR nonsense alleles.

Overall, using the computer-based approach, the authors confirmed most of the previous findings and reassured the community that “the endogenous mRNA NTC contexts are diverse enough to allow accurate prediction of readthrough of PTCs”, which I find important. In addition, they also documented that “the use of a small fragment of native PTC context” is sufficient for studying a SC-RT potential of a given naturally occurring PTC.

Response: We thank the reviewer for finding our work to be important, and for the helpful comments.

Specific comments, suggestions (in the order of their appearance in the text):

- Abstract: “suggestion a tRNA role in readthrough regulation”; did you mean a P-site tRNA? Please specify.

Response: We meant P-site tRNA in that sentence. We have corrected this in the abstract.

- Page 4; “As expected, the negative control ...”; this sentence is confusing.

Response: We have rewritten this sentence and a clarifying additional sentence now precedes it. We believe that these revisions eliminate the confusing aspects of the original version.

- The fact that the untreated model had a low number of mRNAs with detectable SC-RT is, in my opinion, a major, unfortunate setback of this study. For example, even stop codons and nt +4 show low score in Fig. 1 (in contrast to 3' UTR, which is a bit puzzling for me), and predictions from the untreated model do not look very convincing (Fig. 4).

Response: We agree that it is unfortunate that the untreated cells yield lower numbers of mRNAs with detectable readthrough, and this may be why the prediction model did not perform well for untreated cells and the resulting feature importance scores were unexpected. Increasing sequencing depth to increase data richness in future experiments may improve the prediction accuracy. Please keep in mind that we were working with a published data set.

- Page 5, very top, “general trends.” I think it would interesting to pick all genes that are known to allow SC-RT and show whether their stop codon contexts comply or not with these trends.

Response: We explored the readthrough efficiency of 14 genes known to undergo programmed stop codon readthrough (SCR) by experimental validation (compiled in a recent review by Manjunath et al. 2022), many of which have a readthrough motif of UGACUAG. We determined whether these known SCR mRNAs have higher readthrough efficiency in our ribosome profiling data set. While not all SCR mRNAs are expressed or have sufficient expression for their readthrough to be detected at our sequencing depth, those that can be detected tend to have higher readthrough efficiency than average (Fig. 2A, below, red and green). Since 13 out of 14 readthrough genes use a UGA stop codon in their mRNAs, we compared readthrough of these mRNAs against the distribution of readthrough efficiency for all UGA-containing mRNAs and reached a similar conclusion (Fig. 2B, below, red and green). These observations are consistent with the association of the UGAC quadruplet with high readthrough. However, mRNAs having the UGACUAG motif but have not been shown to undergo readthrough experimentally also exist and some of them also exhibit higher readthrough in this ribosome profiling data (Fig. 2, below, blue). Thus, other mRNA features also play a role in regulating readthrough efficiency. We have included this analysis in our revised manuscript as the new Extended Data Fig. 2 and discuss it on p. 5 of the Results.

Figure 2. Readthrough efficiency of mRNAs known to undergo programmed stop codon readthrough (SCR mRNAs, red and green) listed in Manjunath et al. 2022 relative to the distribution of readthrough efficiency of all mRNAs in the sample (A) or UGA-containing mRNAs (B). For comparison, non-SCR mRNAs that have the UGA CUAG motif, which appears in many SCR mRNAs, are also plotted (blue). The gene names are labeled. Mean and median readthrough efficiency are indicated by brown and grey dashed lines, respectively (They are very close and superimposed in most cases).

- Figure 2D; I think the striking case to comment on is ACG, which worked well even in untreated but for some reason was overlooked. None of the other three Thr codons showed such specificity.

Response: The difference in median readthrough efficiency between mRNAs with an ACG codon in the P-site compared to the sample median may seem striking, but it did not pass the statistical test due to low sample size. Since we had to split the data into 60 groups for 60 codons, each group would average about $685/60 \approx 11$ mRNAs ($n = 11$) for the “Untreated” condition. Actually, there are only three mRNAs with ACG in the Untreated condition ($n = 3$) while there are 22 mRNAs with ACG in the G418-treated condition (0.5 ug) ($n = 22$). In the latter set with a bigger sample size we see that the apparent difference is gone (Fig. 2D, ACG tile is no longer red).

- As for the 3' UTR effect, have the authors considered analyzing the role of prospective sec. structures immediately following stops (mentioned in the intro)?

Response: To address the reviewer’s concern regarding whether the 3'-UTR effect is related to secondary structures, we carried out additional analyses where we used an established mRNA secondary structure prediction tool, RNALfold of the ViennaRNA package (Lorenz et al. 2011) and calculated pairwise Spearman’s correlation coefficient between the minimum free energy (MFE) of the most stable predicted structure for a given 3'-UTR sequence (lowest MFE), readthrough efficiency, and 3'-UTR length (Fig. 3, below). The majority of 3'-UTRs (except for 7 across all samples) have at least one predicted structure. We observed that, in all samples, the relationship between readthrough efficiency and MFE is much weaker than that between readthrough efficiency and 3'-UTR length; in some cases, there is no correlation. However, the strongest correlation is the 3'-UTR length vs. MFE; the negative correlation indicates that the longer the 3'-UTR is, the lower the MFE is (i.e., the stronger the structure is). These results indicate that stronger secondary structures tend to occur in longer 3'-UTRs because longer sequences generally have a higher chance of forming stable structures, and that these structures, if they really did occur in the samples, did not account for the 3'-UTR effect observed in the data. We have included these analyses in the revised manuscript as a new Extended Data Fig. 5A and pp. 8 of the Results.

Figure 3. Correlation matrix showing pairwise Spearman's correlation coefficient for readthrough efficiency vs. 3'-UTR length, readthrough efficiency vs. minimum free energy (MFE) in the 3'-UTR, and 3'-UTR length vs. MFE for each sample.

- *Dual Luc measurements.* They are mentioned in the article, but they are not actually shown (just raw measurements as appended xls files), nor is there any reference to them in the text (where to find them). This should be corrected.

Response: As noted above in our responses to Reviewer #1, the Dual Luc raw measurements and calculated ratios were both provided in Supplementary Table 2 and the ratios were also provided on the GitHub repository. An updated version, now designated Extended Data Table 1 is included with the revised manuscript and it is now cited in the Methods section of the revised manuscript (pp. 17-18). As noted above, we have also appended an illustrative file for the reviewers (**Examples of Calculations 1-4-24**) that depicts our results and methods for determining readthrough activity.

- *E585X seems to be even a bigger outlier than W822X, at least in untreated. Why was it ignored?*

Response: The apparent outlier status of E585X in the Untreated condition is due to the very low readthrough efficiency predicted for E585X (i.e., E585X is farther from others along the y-axis but not x-axis). However, as noted above, the predictions in "Untreated" cells are not as accurate as those of drug-treated cells so we cannot determine whether this outlier is a real biological outlier or not.

- *Does it really make sense to consider the impact of the 3' UTR length when it comes to PTCs (Ext. Data Fig. 4)?*

Response: Please refer to our response to Reviewer #3 below.

- Discussion; I would be interested to hear the author's thoughts on the mechanistic contribution of +5 and +9 bases (mentioned on page 11) to SC-RT.

Response: The role of the +5 base is most likely through its stacking interaction with C₁₆₉₈ of 18S rRNA that protrudes into the mRNA channel, as shown by cryo-EM (Shao et al. 2016). As with the +4 base, this +5 base interaction is more stable with purines, although unlike the +4 base where purines are readthrough-inhibiting, A at +5 is readthrough-promoting. For the +9 base, since it still resides in the ribosome's mRNA channel (inferred from its protection from RNaseI digestion in ribosome profiling experiments and from structural studies (Ingolia et al. 2009; Jenner et al. 2010; Ben-Shem et al. 2011; Cridge et al. 2018; Tate et al. 2018), we think a similar scenario may apply where specific interactions between the +9 base and an rRNA base or amino acid residue(s) of a ribosomal protein are more conducive to a termination reaction or less conducive to an elongation reaction. The latter scenario was shown recently by Biziaev and colleagues for multiple 3' nucleotide contexts (Biziaev et al. 2022). We have added a paragraph detailing this information in the Discussion (pp. 12 of the revised manuscript).

- Page 12, top paragraph. Our and O. Namy's labs have also contributed quite extensively to elucidation of the "base-pair kinetics" in yeast and mammalian cells, in other words, to deciphering which nc-tRNAs do and which don't promote SC-RT and why, so I humbly ask the authors to cite at least some of these studies. This is indeed just a comment, not a requirement.

Response: We apologize for this oversight and now include references to your work and Namy's in the pertinent section of the Discussion (p. 13 of the revised manuscript).

Thank you for giving me the opportunity to review this article. Leos S. Valasek

Reviewer #3 (Remarks to the Author):

In this manuscript, Mangkalaphiban and colleagues apply a machine learning approach to reanalyze ribosome profiling data from HEK293 cells to assess readthrough efficiency. They verified the impact of features such as the termination codon identity and the importance of the fourth nucleotide and suggested a role of long 3'UTRs in enhancing readthrough. They also suggest the adoption of machine learning in readthrough prediction, which is also an interesting application.

Response: We thank the reviewer for considering that our application of machine learning to readthrough is interesting.

1. Results, page 7: "Readthrough efficiency increases with 3'UTR length": The p values of Figure 3 are too low to allow such a bold statement. Moreover, there is thin evidence in mammalian systems that PABPC plays a role in efficient translation termination based on an *in vitro* translation system under limiting release factor concentrations.

Response: While the p values are low, they are significant, suggesting that 3'-UTR length plays a role in at least some mRNAs. This analysis is also not controlled for

other more important features, such as the stop codon and nt +4 identities, so their roles potentially “muddle” the correlation. To address this concern, we calculated ρ of readthrough efficiency vs. 3'-UTR length for each mRNA group based on their stop codon and nt +4 identities (Fig. 4, below). We found that the correlation is as high as 0.39 for mRNAs with specific stop codon context, such as UAAG in gentamicin-treated cells and UAGU in tobramycin-treated cells, while the correlation is close to zero in some other cases. These results demonstrate that the 3'-UTR effect (and potentially PABP's involvement) may only matter or be observable in some cases, depending on stop codon, nt +4, and aminoglycoside treatment. Although these results are not direct evidence of PABP's involvement in termination, they may still help explain why PABP's role in termination has been elusive, as PABP's role or the extent of its role may be specific to a system or an mRNA's context. We have included this analysis on pp. 8 of the revised manuscript and as a new Extended Data Fig. 5B.

Figure 4. Readthrough efficiency vs. 3'-UTR length for all mRNAs (“All”, same as Fig 3 in the manuscript) or mRNAs having certain combinations of stop codon and nt +4 in the sample. Spearman's correlation coefficient (ρ) is represented by the color spectrum as well as labeled. Larger tile size indicates significant correlation (p -value < 0.05) and smaller tile size indicates insignificance (p -value \geq 0.05).

2. Results, page 4: I failed to find in Figure 1 the feature of the long 3'UTRs, and therefore, I cannot assess the corresponding readthrough efficiency. Please clarify which part of the figure describes the effect of the 3'UTR length.

Response: Since Fig. 1 in the manuscript only represents feature importance scores from the random forest models, it only indicates whether 3'-UTR length is important as a predictor of readthrough efficiency (the intensity of the red color) in light of other features, but it does not explain how 3'-UTR length influences readthrough efficiency. The relationship between 3'-UTR length and readthrough efficiency is then described in Fig. 3 of the manuscript.

Considering points 1 and 2, I cannot conclude a clear correlation between the 3'UTR length and readthrough efficiency, which is the only novel feature that this analysis proposes.

Response: As we discussed in the above two points, 3'-UTR length may play a role in readthrough efficiency regulation, but it is not the major one, not as much as the stop codon and nt +4. Moreover, as illustrated in Fig. 4 (above), the 3'-UTR effect appears

to be dependent on other factors. So, a high correlation between 3'-UTR length and readthrough efficiency is not expected and its general role in PTC readthrough prediction is negligible.

3. *There are increasing concerns in the translation community about using bicistronic reporter assays to assess translation efficiency, especially based on DNA transfections which can additionally lead to transcription/splicing aberrancies. There is also evidence that test sequences placed between two reporter genes can inadvertently modify the expression or functionality of either reporter. This can result in misleading outcomes, potentially leading to inaccurate findings published in the scientific literature (for instance, <https://doi.org/10.1073/pnas.2122170119>). For this reason, given that Dual luciferase assays are the only additional experimental input in this manuscript (Fig.4), it is suggested at least to perform RNA transfections instead of DNA-based assays.*

Response: We appreciate the reviewer identifying a significant gap in our explanation of the luciferase reporters that were used to measure readthrough of CF-associated PTCs. The dual luciferase readthrough reporters used in our manuscript are monocistronic, with the translational start site located at the beginning of the *Renilla* luciferase ORF, which is used as a normalization control to correct for differences in transfection efficiency, translation initiation, and mRNA abundance. The *Renilla* ORF is followed downstream by an in-frame readthrough cassette containing a PTC flanked by 3 codons of natural upstream and downstream *CFTR* mRNA context followed downstream by an in-frame firefly luciferase. Thus, the constructs are identical with the exception of the 21 nucleotides located in the readthrough cassette that represent the unique sequence contexts examined (3 codons upstream, the PTC, and the 3 codons downstream). In our readthrough efficiency calculations, relative readthrough is expressed as (firefly units / *Renilla* units) x100. We have added these key points to the Materials & Methods section of the revised manuscript (p. 17).

To confirm that the constructs are expressed as expected, we have now performed western blotting, using *Renilla* and firefly antibodies (Fig. 5, next two pages, and new Extended Data Fig. 6 in the revised manuscript) using lysates from HEK293 cells transfected with each reporter construct. If termination is efficient at the PTC, we expect only the 37kDa *Renilla* luciferase protein to be expressed. If termination is suppressed at the PTC, then a 100 kDa fusion protein consisting of *Renilla* luciferase fused to firefly luciferase would be produced. Note that the *Renilla* antibody most often detected a strong 37 kDa band and a much weaker 100 kDa band. The firefly antibody detected the full-length 100kDa band more readily than the *Renilla* antibody, which was less sensitive. This western blotting approach was used to determine: a) whether a 37kDa premature termination product could be identified in all extracts derived from constructs harboring PTCs, b) whether a 100kDa readthrough band could be identified in extracts of the cells manifesting readthrough activity, and c) whether any aberrant bands were detected that might raise concerns about the basis of luciferase activity observed with a given extract. As now reported in the Results and Materials and Methods of the revised manuscript (pp. 9 and 17) and in Fig. 5 (below), we find that all extracts of cells expressing PTC-containing constructs manifest the 37kDa premature termination product, almost all expressed the 100kDa readthrough protein, and only the two S434X alleles (UGA and UAA) expressed aberrant polypeptides. The lack of detectable 100kDa bands in W1282X (UGA), Y122X

(UAA), and Y1092X (UAA) extracts (even with G418 treatment) most likely reflected the relatively modest readthrough activity in those extracts, but was not of concern because the extracts lacked any unexpected bands. However, the presence of a 70kDa aberrant band in extracts of the two S434X alleles (UGA and UAA) was considered sufficient evidence to exclude those two alleles from further analysis because it was possible that this aberrant band may have firefly luciferase activity. Altogether, these results suggest that the dual luciferase readthrough reporters were not subject to extensive transcription or splicing aberrations. Further, any alternative translational initiation events at an internal AUG would occur whether the constructs were introduced into cells as DNA or RNA transfections.

◀ Truncated Protein: ~37kDa

◀ Full Length Protein: ~100kDa

◀ Tubulin (control): ~55kDa

◀ Spurious Product

Figure 5. Western blotting validation of truncated (~37kDa) and full-length readthrough (~100kDa) products of Dual-Luc constructs from cell lysates using anti-Renilla antibody (left) or anti-firefly antibody (right). UT = untransfected control, WT = G542 WT. Spurious products that may result in luciferase activity are boxed in orange.

4. *Results, page 5: "Readthrough-promoting stop codon and nt+4 occur together less often than expected": The title, as well as the text of this part of the text, was hard to comprehend. The last paragraph provides an overview of the finding is much more clear than the rest of the text.*

- *It is suggested to clarify further what the authors mean by the phrase "allow chances for protein adaptation". Some sentences are too long "To statistically determine... stop codons", which hinders comprehension.*

- *The phrase "For the reference and G-418... over-represented" was not clear to me. What does it mean that their association is truly over-represented?*

Response: We have rewritten this section (and its title) with the goal of eliminating all ambiguities. We hope that the reviewer finds the new version an improvement and easier to read.

REVIEWERS' COMMENTS

Reviewer #1 (Remarks to the Author):

The authors have addressed all of the concerns that I raised. However, I noticed one error in extended data Figure 2. Readthrough on all human UGA CUAG motifs have been experimentally verified by reporter assay (doi:10.1074/jbc.M117.818526). Therefore PHF10 and CGGBP1 should not be in the blue 'Non-SCR mRNAs: UGA CUAG motif'

Reviewer #2 (Remarks to the Author):

The authors addressed all of my questions.

Reviewer #3 (Remarks to the Author):

The authors addressed all my concerns and I support publication of the manuscript in its current form.

Mangkalaphiban et al. 2024: Point by point responses to reviewers' comments

Reviewer #1 (Remarks to the Author):

The authors have addressed all of the concerns that I raised. However, I noticed one error in extended data Figure 2. Readthrough on all human UGA CUAG motifs have been experimentally verified by reporter assay (doi:10.1074/jbc.M117.818526). Therefore PHF10 and CGGBP1 should not be in the blue 'Non-SCR mRNAs: UGA CUAG motif'.

Response: The reviewer raised a useful point about Supplementary Fig. 2 (formerly Extended Data Fig. 2). The reference cited did indeed demonstrate that readthrough of the PHF19 (the reviewer mislabeled it as PHF10) and CGGBP1 mRNAs has been experimentally verified. Accordingly, both mRNAs are now depicted in red in Supplementary Fig. 2.

Reviewer #2 (Remarks to the Author):

The authors addressed all of my questions.

Response: Thanks for taking the time to review our paper.

Reviewer #3 (Remarks to the Author):

The authors addressed all my concerns and I support publication of the manuscript in its current form.

Response: Thanks for taking the time to review our paper.